# Fair Text Classification with Wasserstein Independence

Thibaud Leteno[1*]   Antoine Gourru[1*]   Charlotte Laclau[2]
Rémi Emonet[1]   Christophe Gravier[1]

Laboratoire Hubert Curien, UMR CNRS 5516, Saint-Etienne, France
Télécom Paris, Institut Polytechnique de Paris, Paris, France
{thibaud.leteno, antoine.gourru, remi.emonet, christophe.gravier}@univ-st-etienne.fr
charlotte.laclau@telecom-paris.fr

## Abstract

Group fairness is a central research topic in text classification, where reaching fair treatment between sensitive groups (e.g. women vs. men) remains an open challenge. This paper presents a novel method for mitigating biases in neural text classification, agnostic to the model architecture. Considering the difficulty to distinguish fair from unfair information in a text encoder, we take inspiration from adversarial training to induce *Wasserstein independence* between representations learned to predict our target label and the ones learned to predict some sensitive attribute. Our approach provides two significant advantages. Firstly, it does not require annotations of sensitive attributes in both testing and training data. This is more suitable for real-life scenarios compared to existing methods that require annotations of sensitive attributes at train time. Secondly, our approach exhibits a comparable or better fairness-accuracy trade-off compared to existing methods. Our implementation is available on Github[1].

## 1  Introduction

Machine learning algorithms have become increasingly influential in decision-making processes that significantly impact our daily lives. One of the major challenges that has emerged in research, both academic and industrial, concerns the fairness of these models, i.e. their ability to prevent predictions related to individuals to be based on sensitive attributes such as gender or ethnicity. In this article, we focus on the problem of fairness in the domain of Natural Language Processing (NLP) (Bender et al., 2021; Osoba and Welser IV, 2017; Schwemmer et al., 2020). While many studies already report biases in NLP systems (Sun et al., 2019; Hutchinson et al., 2020; Tan and Celis, 2019;

---

*Equal contribution.
[1]https://github.com/LetenoThibaud/wasserstein_
fair_classification

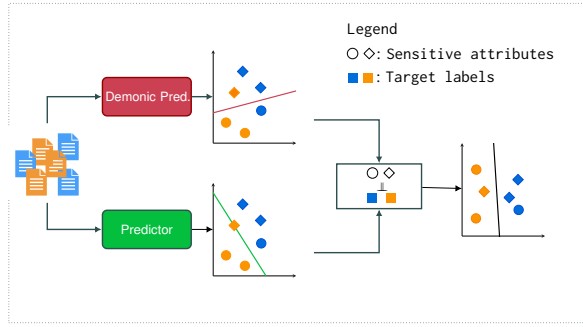

Figure 1: Our method, WFC, modifies the representation space of documents such that it is fairer when training a classifier on top. To do that, it makes it independent of a *"demonic"* model that predicts the sensitive attribute.

Liang et al., 2021; Bender et al., 2021), these issues become even more significant with the advent of public-ready AI-powered NLP systems such as ChatGPT (OpenAI) or Google Bard (Pichai), making the need for fair NLP solutions even more compelling. As more researchers work to overcome these shortcomings, the first problem is to define what *fairness* is. Such a definition may hardly be consensual or is at least difficult to establish, as it depends on situational and cultural contexts (Fiske, 2017). In this work, we adopt the most common definition of group fairness, and the one adopted by laws in several countries, which is based on the notion of disparate impact: a prediction model is considered to have a disparate impact if its results disproportionately harm (or benefit) people with certain sensitive attribute values (e.g., women, black people).

In this work, we focus on group fairness for neural text classification as it is one of the most ubiquitous tasks in our society, with prominent examples in medical and legal domains (Demner-Fushman et al., 2009) or human resources (Jatobá et al., 2019), to name a few. Neural text classification relies on text encoders, which are parameterized and learned functions that map tokens (arbitrary

text chunks) into a latent space of controllable dimension, usually followed by a classification layer. Built upon the Transformers architecture (Vaswani et al., 2017), popular Pre-trained Language Models (PLMs) such as BERT (Devlin et al., 2019), GPT3 (Radford et al., 2019) or Llama (Touvron et al., 2023) leverage self-supervised learning to train the text encoder parameters. In modern NLP pipelines, these PLMs are further fine-tuned on the supervised task at hand. Ultimately, PLMs accumulate uncontrolled levels of unfairness due to unbalanced learning data or algorithmic biases, for instance. This results in observable biases in predictions but also in the latent space as studied in (Zhao et al., 2019; May et al., 2019).

We propose a novel approach (see Figure 1), to mitigate bias in text encoders, that aims to tackle bias directly in the latent space on which documents are projected, thus making our model applicable to any text encoder or decoder (e.g. BERT or LLAMA). To proceed, we disentangle the neural signals encoding bias from the neural signals used for prediction. The proposed architecture is based on three components. First, two Multi-Layer Perceptrons (MLPs): the first one whose objective is to predict the sensitive attribute, and the second one is dedicated to the prediction task at hand. Then, a third MLP, referred to as a critic, approximates the Wasserstein distance that acts as a regularizer in our objective function. Our proposition overcomes a major shortcoming of prior studies: they rely on the availability of the sensitive attributes at train time. A constraint that is incompatible with recent regulations as the new European ones, that enforce more stringent requirements for the collection and utilization of protected attributes. Prior studies are thus more difficult to use in practical settings. In the following, we will show that our approach can address this limitation by avoiding the use of this information during both testing and training.

The rest of this paper is organized as follows. Section 2 presents recent advances regarding fairness in NLP. Section 3 argues about our motivation and provides the background knowledge to understand our contribution. Section 4 proceeds with the description of the proposed approach, its theoretical analysis, and algorithmic implementation. Section 5 introduces the setting of our experiments, and Section 6 presents the results and interpretation. The last section concludes the paper and gives a couple of hints for possible future research.

## 2 Related Works

Numerous studies have been conducted on how to tackle bias in machine learning systems. Approaches to reinforce fairness can be divided between pre-, in-, and post-processing methods. In the NLP literature, common pre-processing techniques consist of data rebalancing (Park et al., 2018) or embedding debiasing (Wang et al., 2020; Bolukbasi et al., 2016). Yet, despite the efficiency of those methods, Kaneko et al. (2022) and Tokpo et al. (2023) showed that other biases can be learned during the training or fine-tuning processes. On the other hand, post-processing procedures aim at correcting biases after the learning step, through model calibration (Zhao et al., 2017; Jia et al., 2020) or data projection (Ravfogel et al., 2020) (INLP). We refer to (Sun et al., 2019; Blodgett et al., 2020) for more exhaustive surveys of bias in NLP.

Recently, adversarial methods (Beutel et al., 2017; Zhang et al., 2018; Elazar and Goldberg, 2018) have been investigated to mitigate biases. Han et al. (2021c,b) respectively suggest using several discriminators where each learns different hidden representations and applying an adversarial approach in- and cross-domain to train the adversary on a different dataset, with methods called Adv and GATE. Differently, recent contributions focus on directly constraining the objective to improve fairness (Shen et al., 2022b; Han et al., 2021a). For instance, by adding some fairness metric, such as the Equal opportunity that we will define later in this paper, to the objective function.

Our approach is at the crossroad of these two philosophies: on the one hand, we propose to train a biased model whose sole purpose is to predict the sensitive attribute and use this latter to enforce fairness in our main prediction model. On the second hand, we minimize a classifier loss with a regularization term measuring the dependence between the latent representations of the classifier and some *unfair* representations, using Wasserstein distance. While many works use the Kullback–Leibler (KL) divergence to measure the mutual information between representations, Ozair et al. (2019) show several limitations: the KL-divergence is sensitive to small perturbations in the data, and exploiting it for estimating the mutual information does not scale up properly. Thus, they suggest an improvement thanks to the Wasserstein distance. Other methods based on this distance suggest focusing on the distance between the distributions of predictions to en-

force fairness (Jiang et al., 2020; Risser et al., 2022). Finally, most approaches aforementioned depend on the availability of sensitive attribute annotations in the training data, and as Kenfack et al. (2023) recently emphasized, employing proxy-sensitive attributes often worsens the fairness-accuracy trade-off. They also propose a proxy model to retrieve the missing sensitive attributes, adapted to improve the model's fairness.

**Limits of existing approaches and positioning** Compared to the adversarial approaches previously mentioned, ours is conceptually closer to (Nam et al., 2020), while their methodology is conducted on images rather than textual data. We also distinguish from their research by the use of the Wasserstein distance to evaluate the mutual information between the two models' representations instead of focusing the learning of the main model on samples going against the prejudice of the biased network. Like Ozair et al. (2019), we exploit the Wasserstein distance as an estimator of the mutual information. However, while they use it to measure mutual information to improve representational learning on images, we consider sensitive attributes in the mutual information estimation and use it to improve the model fairness. Our proposition is related to Risser et al. (2022), yet, we do not use the model outputs directly as we use hidden representations, of fixed-size and task-independent dimensions, of Pre-trained Language Models that encode information on sensitive attributes. Additionally, we do not use the Wasserstein distance to compute the distance between each group's prediction probability but to enforce the independence of the representation from unfair representations. By using those latent representations in the Wasserstein-regularization term, the model is encouraged not to encode the sensitive information in the representation during inference. Similarly, in the field of NLP, a related approach is proposed by Cheng et al. (2021). Their method maximizes the mutual information between pairs of sentence representations and their augmented versions, which vary based on the sensitive attribute. These representations go through the same encoder, ensuring that the input is independent of the sensitive information. However, this does not ensure independence between the prediction and the sensitive attribute (Shen et al., 2022a; Cabello et al., 2023). In contrast, our theoretically grounded approach minimizes the mutual information between representations of the same sentence

processed by two different encoders to make the predictions independent of the sensitive attributes. Moreover, their approach depends on identifying biased attribute words, limiting its applicability to cases where substitute words are accessible. This is a constraint we avoid. Lastly, while previous methods primarily targeted classification issues in images or categorical and numerical data, we introduce an approach well-suited for Natural Language Processing. It can be applied to less-explored scenarios, including continuous sensitive attributes and regression tasks.

## 3 Preliminaries

In this section, we introduce the notations used throughout this paper. We also present the definitions of the necessary fairness metrics, and the main concepts, mostly related to the Wasserstein distance which are essential for understanding the rest of the paper.

### 3.1 Motivation

We consider a corpus of $n$ triplets $\{(x_i, y_i, s_i)\}_{i=1}^n$, where $x_i \in \mathcal{X}$ is a short document or a sentence, $y_i \in \mathcal{Y}$ is a label and $s_i \in \mathcal{S}$ corresponds to one or multiple variables, referred to as *sensitive* attributes, such as gender, ethnicity or age. Let us consider binary classification for illustrative purposes. The goal is to predict the outcomes $y_i$ by estimating the conditional probability $p(y = 1|x = x_i)$ through a scoring function $f : \mathcal{X} \to \{0, 1\}$. The prediction associated with $f$ is noted $\hat{y}$. For example, in the context of a social network, a classifier can use the descriptors of a message (e.g., a bag of word representation), $x_i$, to predict whether a message is toxic or not, leading to the decision to ban the message and/or the user who wrote it from the social platform.

### 3.2 Measuring Fairness

In this context, of particular relevance is group-based fairness, which examines how well outcome ($\hat{y}$) consistency is preserved across sensitive groups ($s$). Returning to our example, when determining whether a message is toxic, fairness here implies that the decision is consistent for all users, regardless of gender or ethnicity.

A commonly used group-based metric used to quantify the *(un)fairness* of a given classifier is the Equality of Opportunity **EO** (Hardt et al., 2016) that is satisfied if the prediction made by the clas-

sifier is conditionally independent of the protected attribute, given that the true value is positive (e.g. $y = 1$). In effect, it means that the same proportion of each group receives a *positive* outcome. For binary sensitive attribute ($s \in \{a, \bar{a}\}$) and multi-class objectives, the consensual way of aggregating EO score over classes is the GAP score (De-Arteaga et al., 2019; Ravfogel et al., 2020) defined as follows

$$\mathbf{GAP} = \sqrt{\frac{1}{|\mathcal{C}|} \sum_{c \in \mathcal{C}} (EO_c)^2}, \qquad (1)$$

where the EO for a given class $c \in \mathcal{C}$ is defined by

$$\begin{aligned} \mathbf{EO}_c = & p(\hat{y} = c | y = c, s = a) \\ & - p(\hat{y} = c | y = c, s = \bar{a}). \end{aligned} \qquad (2)$$

Additionally, as fairness often requires determining a trade-off such that reaching equity does not degrade the general classification performance, Han et al. (2021a) proposed the Distance To Optimum (**DTO**) score. This latter measures the accuracy-fairness trade-off by computing the Euclidean distance from a model to an *Utopia point* (point corresponding to the best Accuracy and best Fairness values across all the baselines). The goal is to minimize the DTO.

Finally, we consider the **Leakage** metric that corresponds to the accuracy of a classification model trained to predict the sensitive attribute from the latent representations. It measures the fairness *of the latent representations themselves* and demonstrates unfairness when close to 100 of accuracy.

### 3.3 Fairness as Mutual Information minimization

Mutual Information (MI) is an information-theory-based metric meant to measure the statistical dependence or the amount of information shared between two variables. In our context, given the class of a document predicted by our model along with the value of the sensitive attribute of the document, one can use MI to evaluate the strength of their relationship. Formally, the mutual information is defined as the Kullback-Leibler (KL) divergence between the joint distribution $p(x, y)$ and the product of the marginal distributions:

$$\mathrm{MI}(x, y) = \mathrm{KL}(p(x, y) \| p(x) p(y)). \qquad (3)$$

Fairness can therefore be cast as MI minimization between $\hat{y}$, our prediction (conditioned on $y$,

the ground-truth or not), and $s$, the sensitive attribute, as it will make the prediction and the sensitive attribute less and less dependent. Nevertheless, MI is generally intractable for most real-life scenarios and has strong theoretical limitations as outlined by Ozair et al. (2019). Notably, it requires a number of samples exponential in the value of the Mutual Information to build a high confidence lower bound and it is sensitive to small perturbations in the data sample. Consequently, Ozair et al. (2019) proposed a theoretically sound dependency measure, the *Wasserstein Dependency Measure*, based on Wasserstein-1 distance :

$$\mathrm{MI}_W(x, y) = W_1(p(x, y), p(x) p(y)). \qquad (4)$$

A key feature of the Wasserstein distance is that it can act as a smooth objective function, as shown in the WGAN approach (Arjovsky et al., 2017). More precisely, the Kantorovich-Rubinstein duality expresses $W_1(p(x, y), p(x) p(y))$ as :

$$\begin{aligned} \sup_{\|C\|_L \leq 1} & \mathbb{E}_{x, y \sim p(x, y)}[C(x, y)] \\ & - \mathbb{E}_{x \sim p(x), y \sim p(y)}[C(x, y)], \end{aligned} \qquad (5)$$

where $\|C\|_L \leq 1$ is the set of all 1-Lipschitz functions. Arjovsky et al. (2017) propose to approximate this measure by using a parameterized function, defined as follows :

$$\begin{aligned} \max_{\omega, \|C_w\|_L \leq 1} & \mathbb{E}_{x, y \sim p(x, y)}[C_\omega(x, y)] \\ & - \mathbb{E}_{x \sim p(x), y \sim p(y)}[C_\omega(x, y)], \end{aligned} \qquad (6)$$

where $C_\omega$ is called the critic and is usually a neural network. Wasserstein distance has been efficiently used in many machine learning fields (Frogner et al., 2015; Courty et al., 2014; Torres et al., 2021) and a particularly interesting application is that of fair machine learning (Jiang et al., 2020; Silvia et al., 2020; Gordaliza et al., 2019; Laclau et al., 2021). See Appendix A.1 for further theoretical details on the Wasserstein Distance. The role of this measure in our contribution is detailed in the subsequent sections.

## 4 Our contribution

We are now ready to show how one can cast the problem of group fairness as an independence constraint in the intermediate latent space of the MLPs and derive a theoretically sound approach based on the Wasserstein distance to solve it.

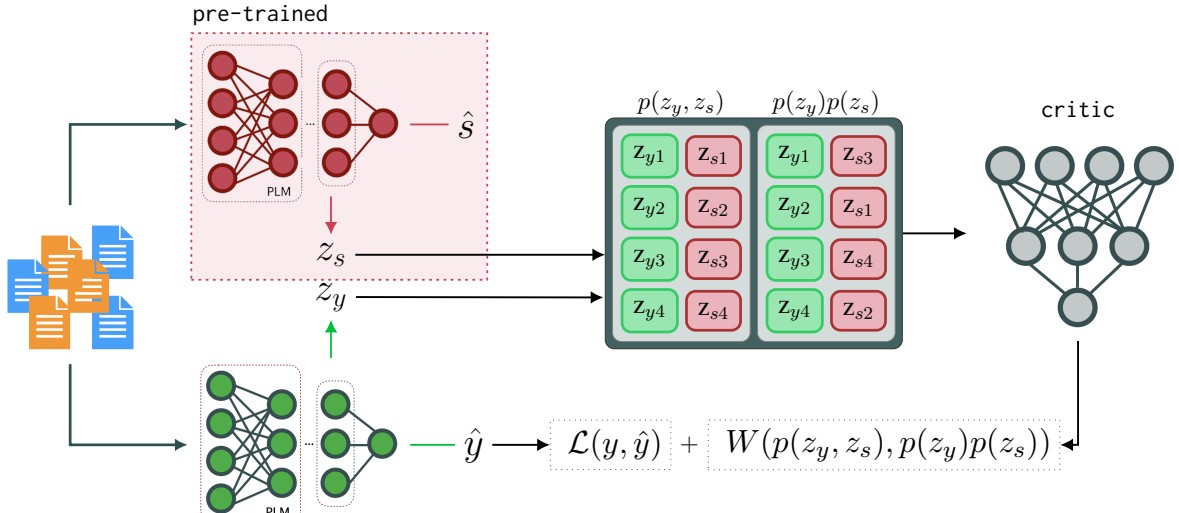

Figure 2: Architecture for a batch of size 4 at train time. The data representation on the left shows how we create dependency or independence between $z_y$ and $z_s$. At inference, only the trained classifier (green) is kept to predict $\hat{y}$.

## 4.1 Definition of the Objective Function

In most recent NLP applications, deep classification is performed as a two-step approach: the scoring function $f$ is a composition of two parameterized functions such that $f = g \circ h$, where $g(x) = z \in \mathbb{R}^d$ projects $x$ in low dimensional space and $h$ is a simple, usually one linear layer neural network with softmax activation followed by an argmax referred to as a classification layer. The objective of $g$ is to produce an embedding $z$ linearly separable with respect to the target of interest. For the deep model predicting $y$, we write $f_c = g_c \circ h_c = h_c(z_c)$, where the index $c$ stands for classification.

As stated earlier, fairness can be defined as minimizing $\mathrm{MI}(\hat{y}, s)$. As $s$ is neither observable nor allowed to be observed in most real-life scenarios, we make use of a surrogate for $s$ that we call the *demonic model*. This deep model is a composition of $f_s = g_s \circ h_s = h_s(z_s)$, where $s$ stands for sensitive attribute. Therefore, in the absence of the sensitive attribute, we can use :

$$\min \ \mathrm{MI}(\hat{y}, \hat{s}). \qquad (7)$$

As the argmax operation producing the hard predictions following the classification layer is not differentiable, we propose to minimize the MI between the latent representations instead of the network final output, leading to optimizing an upper bound of the latter equation (see details in Appendix):

$$\min \ \mathrm{MI}(z_y, z_s) \geq \ \mathrm{MI}(\hat{y}, \hat{s}). \qquad (8)$$

Lastly, we replace MI with $\mathrm{MI}_W$ for the reasons explained earlier. We propose to optimize simultaneously the Wasserstein dependency measure and the more traditional classification loss (e.g. the cross-entropy). Our objective function writes as follow

$$\begin{aligned} \arg \min \ & \mathcal{L}(y, h(z_y)) \\ & + \beta \ W_1(p(z_y, z_s), p(z_y)p(z_s)), \end{aligned} \qquad (9)$$

where $\mathcal{L}$ is the loss function aiming at maximizing the accuracy of the model for predicting $y$ while the role of the second term is to constrain the encoder part of the language model to learn fair representations. The hyper-parameter $\beta \in \mathbb{R}^+$ is a weight allowing some control on the impact of the penalty as the speed of convergence of the two subobjectives may be different. In the following, we refer to the approach minimizing Equation 9 as WFC for Wasserstein Fair Classification (WFC).

## 4.2 Optimization of WFC

The overall architecture of WFC is presented in Figure 2. Given a batch of documents along with their sensitive attribute, we start by generating a representation of each document using a PLM. Then, taking these vectors as input, we train two MLPs to predict $s$ and $y$, respectively. The former is referred to as the *demonic* model in the remained of this paper. Now, assuming that the MLPs consist of

one hidden layer, for the sake of simplicity, we can extract two embedding vectors for all documents, denoted by $z_s$ and $z_y$. Note that the prediction $\hat{y}$ made by the second MLP (in green in Figure 2) can be directly used to compute the first part of our objective function (see Equation 9).

Now for the second term of the loss, which is given by $W_1(p(z_y, z_s), p(z_y)p(z_s))$, we use the approximation proposed by Arjovsky et al. (2017):

$$
\max_{\omega, ||C_w||_L \leq 1} \mathbb{E}_{z_y, z_s \sim p(z_y, z_s)}[C_\omega(z_y, z_s)]
$$
$$
- \mathbb{E}_{z_y \sim p(z_y), z_s \sim p(z_s)}[C_\omega(z_y, z_s)]. \quad (10)
$$

In this context, $C_\omega$ is a MLP, referred to as the `critic` in Figure 2. To enforce the Lipschitz constraint, we clamp the weights to given values ($[-0.01, 0.01]$) at each optimization step[2]. For a batch of documents, the `critic` takes as input the concatenation of $z_y$ and $z_s$ on the one hand, and the concatenation of $z_y$ and $z_s$ randomly drawn from the dataset (equivalent to $z_y \sim p(z_y), z_s \sim p(z_s)$), on the other hand. We then follow the training procedure introduced by Arjovsky et al. (2017) which alternate maximizing Equation 10 in the `critic` parameters for $n_c$ iterations and minimizing Equation 9 for $n_d$ iterations in the $f_y$ classifier parameters. The overview of the training process is detailed in the appendix B.3

**Training the *demonic* model** We pre-train the *demonic* model to predict the sensitive attributes and do not update the *demonic* weights during the training phase of the main model. The benefits are twofold. Firstly, unlike previous works (Caton and Haas, 2020), we require only limited access to sensitive attributes label at training and we do not need access to the labeling of sensitive attributes in the inference regime. As a result, WFC is highly compatible with recent regulations (e.g., US Consumer Financial Protection Bureau). Secondly, the *demonic* model can now be trained in a few-shot fashion if some examples of the training set are annotated with sensitive attributes. However, when no sensitive attributes are available in the training set, we replace the training data of the *demonic* part of the architecture with data from another domain (e.g. another dataset) containing sensitive information for the same attribute. For example, for gender,

we can leverage generated datasets, like the EEC dataset (Kiritchenko and Mohammad, 2018). Thus, we transfer this knowledge from one dataset to another, working towards fairness autonomy regardless of the inclusion of sensitive attributes within the data.

## 5 Experimental Protocol

In our experiments, we intensively use the FairLib package (Han et al., 2022), which provides an access to many state-of-the-art methods and datasets.

### 5.1 Dataset

We employ two widely-used datasets to evaluate fairness in the context of text classification, building upon prior research (Ravfogel et al., 2020; Han et al., 2021c; Shen et al., 2022a). Both datasets are readily available in FairLib.

**Bias in Bios (De-Arteaga et al., 2019).** This dataset, referred to as 'Bios dataset' in the rest of the paper, consists of brief biographies associated with occupations (a total of 28) and genders (male or female). As per the partitioning prepared by (Ravfogel et al., 2020), the training, validation, and test sets comprise $257,000$, $40,000$ and $99,000$ samples, respectively.

**Moji (Blodgett et al., 2016).** This dataset contains tweets written in either "Standard American English" (SAE) or "African American English" (AAE), annotated with positive or negative polarity. We use the dataset prepared by (Ravfogel et al., 2020), which includes $100,000$ training examples, $8,000$ validation examples, and $8,000$ test examples. The target variable $y$ represents the polarity, while the protected attribute corresponds to the ethnicity, indicated by the AAE/SAE attribute.

### 5.2 Baselines

Except for the classical cross-entropy loss without fairness constraint (CE) that we run ourselves, we report the results from (Shen et al., 2022a; Han et al., 2022) on these two datasets. The considered baselines are INLP (Ravfogel et al., 2020), the ADV method (Han et al., 2021c), FairBatch (Roh et al., 2021), GATE (Han et al., 2021a) and Con, displaying the $dp$ and $eo$ versions (Shen et al., 2022a).

### 5.3 Evaluation Tasks

For training a vanilla text classification model, we follow the protocol proposed by Han et al. (2022):

---

[2]We also tested some more recent improvements of Lipschitz constraint enforcement (Gulrajani et al., 2017; Wei et al., 2018). Interestingly, all lead to poor performance

a frozen BERT encoder followed by a 3-layer MLP. We use accuracy to assess the classification performance. Fairness for all models is evaluated against three metrics presented earlier: GAP, referred to as "Fairness" in previous works (Han et al., 2022; Shen et al., 2022a), and the Distance To Optimum (DTO) proposed by Han et al. (2021a) (we follow the methodology of Shen et al. (2022a) and evaluate the DTO on the average fairness and accuracy of the best empirical results for each metric over all models to build the utopia point). Finally, we consider the Leakage score.

**Task 1: Fair Classification** We first compare our method against state-of-the-art approaches. We use the representation generated by a base BERT model as an input to the MLPs. For Bios, the *demonic* MLP is trained on 1% of the training set and obtains 99% accuracy for predicting the sensitive attributes on the test set. Similarly, the *demonic* MLP obtains 88.5% accuracy on Moji.

**Task 2: *Demonic* transfer** We conduct these experiments for Bios and train a *demonic* MLP either on the EEC dataset (Kiritchenko and Mohammad, 2018) or the Marked Personas dataset (Cheng et al., 2023). We then evaluate the performance of the *demonic* MLP to predict gender on the Bios test dataset. When training on the EEC dataset we obtain 98.1% of accuracy, and on the Marked Personas dataset, we obtain 98.4% of accuracy. We repeat Task 1, with those variants of the *demonic* MLP. We focus on Bios in this experiment. For Moji, it would require to have access to other datasets with the same protected attribute (ethnicity).

**Task 3: Use of representations from different layers** In the previous experiments, following approaches presented in (Han et al., 2022), the Wasserstein distance is approximated using the last hidden representations of the 3-layer MLP. We compare this approach, on both datasets, with the use of the first hidden representations of the MLP and with the output logits. For the latter, the Wasserstein is estimated between distributions of different dimensions: for example, in the case of Bios, 2 for the *demonic* MLP corresponding to the sensitive attributes and 28 for the classification MLP corresponding to the labels.

**Task 4: Independence with predicted hard sensitive attributes** To evaluate the impact of using

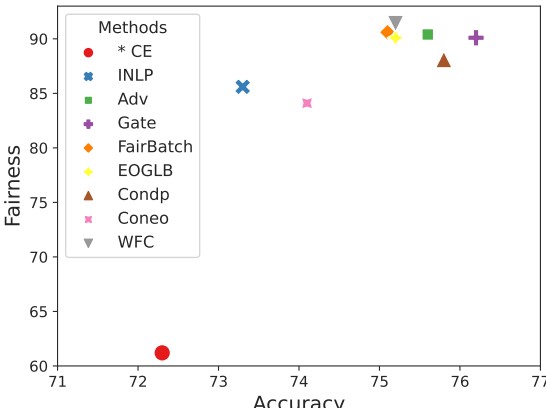

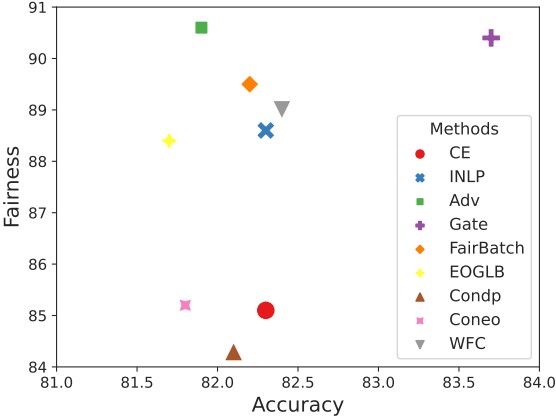

Figure 3: Visualization of the fairness-accuracy trade-off on (a) Moji and (b) Bios. Values correspond to the average results.

the representation $z_s$, we reproduce Task 1, but replace $z_s$ with the sensitive attributes predicted by the *demonic* MLP: $\hat{s}$ and concatenate $z_y$ and $\hat{s}$ dependently and independently when computing the Wasserstein distance. Note that we do not encounter a problem with the non-differentiability for $\hat{y}$ (with the argmax operation as for $\hat{s}$ as mentioned in Section 4.1) since the *demonic* model is pre-trained.

### 5.4 Implementation Details

Our architecture is composed of three components: two classifiers and a critic. The details of the MLPs used to parameterize each component are given in Appendix B. We find the best hyperparameters for our models by grid-search cross-validation over the MLP and Critic learning rates, the value of $n_d$ (number of batches used to train the main MLP), the layers producing $z_s$ and $z_y$, the value of $\beta$ and the value used to clamp the weights to enforce the Lipschitz constraint. The values allowing us

Table 1: Results on Moji (top) and Bios (bottom). For baselines, results are drawn from (Shen et al., 2022a). We report the mean ± standard deviation over 5 runs. * indicates the model without fairness consideration.

| Model | Accuracy ↑ | Fairness ↑ | DTO ↓ | Leakage ↓ |
|---|---|---|---|---|
| *CE | 72.3 ± 0.5 | 61.2 ± 1.4 | 31.0 | 87.9 ± 3.3 |
| INLP | 73.3 ± 0.0 | 85.6 ± 0.0 | 7.02 | 86.7 ± 0.6 |
| Adv | 75.6 ± 0.4 | 90.4 ± 1.1 | 1.71 | 78.8 ± 6.0 |
| Gate | **76.2 ± 0.3** | 90.1 ± 1.5 | 1.90 | 100.0 ± 0.0 |
| FairBatch | 75.1 ± 0.6 | 90.6 ± 0.5 | 1.78 | 88.4 ± 0.4 |
| EOGLB | 75.2 ± 0.2 | 90.1 ± 0.4 | 2.15 | 85.7 ± 1.2 |
| Condp | 75.8 ± 0.3 | 88.1 ± 0.6 | 3.92 | **54.2 ± 0.9** |
| Coneo | 74.1 ± 0.7 | 84.1 ± 3.0 | 8.17 | 80.1 ± 4.2 |
| WFC | 75.2 ± 0.1 | **91.4 ± 0.3** | **1.17** | 86.9 ± 0.2 |
| Model | Accuracy ↑ | Fairness ↑ | DTO ↓ | Leakage ↓ |
| *CE | 82.3 ± 0.2 | 85.1 ± 0.8 | 5.67 | 98.0 ± 0.0 |
| INLP | 82.3 ± 0.0 | 88.6 ± 0.0 | 2.44 | 97.6 ± 0.1 |
| Adv | 81.9 ± 0.2 | **90.6 ± 0.5** | 1.80 | 88.6 ± 4.6 |
| Gate | **83.7 ± 0.2** | 90.4 ± 0.9 | **0.20** | 100.0 ± 0.0 |
| FairBatch | 82.2 ± 0.1 | 89.5 ± 1.3 | 1.86 | 98.0 ± 0.3 |
| EOGLB | 81.7 ± 0.4 | 88.4 ± 1.0 | 2.97 | 97.2 ± 0.5 |
| Condp | 82.1 ± 0.2 | 84.3 ± 0.8 | 6.50 | **76.3 ± 1.5** |
| Coneo | 81.8 ± 0.3 | 85.2 ± 0.4 | 5.72 | 84.9 ± 3.4 |
| WFC | 82.4 ± 0.1 | 89.0 ± 0.3 | 2.06 | 96.5 ± 0.5 |

Table 2: Evaluation of the impact of parameter $\beta$ (Equation 9) on the model's leakage. Results on Moji (top) and Bios (bottom).

| $\beta$ | DTO ↑ | Accuracy ↑ | Fairness ↑ | Leakage ↓ |
|---|---|---|---|---|
| 1 | 1.2 | 75.2 | 91.4 | 86.9 |
| 5 | 4.2 | 72.1 | 93.4 | 81.1 |
| 10 | 5.8 | 70.5 | 92.1 | 81.6 |
| 20 | 7.6 | 68.6 | 92.1 | 84.1 |
| $\beta$ | DTO ↑ | Accuracy ↑ | Fairness ↑ | Leakage ↓ |
| 1 | 2.1 | 82.4 | 89.0 | 96.5 |
| 5 | 2.3 | 81.8 | 89.3 | 87.4 |
| 10 | 2.8 | 81.8 | 88.5 | 88.8 |
| 20 | 3.6 | 81.3 | 88.0 | 81.8 |

to obtain the lower DTO during this process are presented in Appendix B. The architecture details of the MLP for the leakage are provided in (Shen et al., 2022a) as we use the same configuration.

# 6 Results

## 6.1 Task 1: Fair Classification

We compare WFC with text classification baselines. For Moji, (Table 1 and Fig. 3), accuracy of WFC is higher than the accuracy of CE and it is equivalent to competitors. On the fairness metrics, we outperform all other baselines and obtain the best DTO. For Bios (Table 1 and Fig. 3), our method is competitive with the other baselines and ranks 4 out of 9 in terms of accuracy-fairness trade-off. In comparison, with equivalent methods in terms of

DTO (INLP, FairBatch, and Adv), WFC improves either the performance or the fairness. Especially, WFC has the second-best accuracy compared to baselines. Finally, we note that WFC is more stable in terms of Fairness compared with other approaches having on average the best results for this metric (along with a smaller standard deviation). Eventually, even when our method does not outperform the baselines (e.g., Bios dataset), it still exhibits noteworthy properties, particularly its ability to achieve competitive performances without access to the sensitive attributes in the training set. We evaluate this capability in the next subsection. We also explore the ability of our proposition to improve the leakage. We initially aim at improving the fairness while maintaining the accuracy of the model. Yet, our method allows to improve leakage by increasing the value of $\beta$ in Equation 9, in other words, we give more importance to the Wasserstein regularization in the loss. We note in Table 2 that with a higher $\beta$, the leakage decreases. However, on both datasets, the accuracy, that we want to preserve, decreases and the trade-off worsens as we get better for the leakage. To sum up, reducing leakage makes it more challenging to retrieve sensitive attributes but could result in unintended information loss needed for the classification task affecting the performance. Ultimately, we want to enhance fairness while keeping a good performance and this objective may not necessarily match with a strong leakage improvement.

## 6.2 Task 2: Demonic Transfer

For this task, we pre-train the *demonic* model on other datasets. Table 3a shows that we achieve similar results as when the pre-training is done using the same dataset. The average loss of accuracy and fairness are not significant. These results are promising for improving fairness, especially in situations where collecting sensitive data is not feasible or when only partial information is accessible.

## 6.3 Task 3: Use of representations from different layers

On both datasets (Table 3b), accuracy is rather stable regardless of the layers used to compute the Wasserstein distance. Still, the best results are obtained using the last hidden representations. However, while we note a slight decrease in fairness on Bios when using representations from other layers, the decrease becomes much more significant on Moji. Using the last hidden layer is the best option.

Table 3: Summary of the results for tasks 2, 3 and 4. We report the mean $\pm$ standard deviation over 5 runs for all tasks. Boldface numbers are the best results, and a star indicates that the difference is statistically significant according to a signed-rank Wilcoxon test (i.e. with a p-value lower than 0.01).

| Data | Accuracy ↑ | Fairness ↑ | DTO ↓ | Leakage ↓ |
|---|---|---|---|---|
| Bios 1% | **82.4 ± 0.1** | **89.0 ± 0.3** | **2.06** | 96.5 ± 0.5 |
| EEC | 82.2 ± 0.4 | 88.9 ± 0.4 | 2.26 | 97.5 ± 0.3 |
| MP | **82.4 ± 0.3** | 88.9 ± 0.4 | 2.14 | **96.4 ± 0.5** |

(a) Task 2: comparison between several scenarios for training the *demonic* model for prediction on Bios.

| Layer | Accuracy ↑ | Fairness ↑ | DTO ↓ | Leakage ↓ |
|---|---|---|---|---|
| | | **Bios** | | |
| Last hid. | **82.4 ± 0.1*** | **89.0 ± 0.3*** | **2.06*** | 96.5 ± 0.5 |
| First hid. | 81.9 ± 0.2 | 86.7 ± 0.4 | 4.29 | 96.5 ± 0.6 |
| Last lay. | 82.1 ± 0.6 | 87.5 ± 0.3 | 3.49 | **87.0 ± 1.1*** |
| | | **Moji** | | |
| Last hid. | **75.2 ± 0.1*** | **91.4 ± 0.3*** | **1.17*** | 86.9 ± 0.2 |
| First hid. | 74.3 ± 0.1 | 80.8 ± 1.0 | 11.4 | 85.6 ± 0.6 |
| Last lay. | 73.5 ± 0.0 | 70.2 ± 0.2 | 21.9 | **64.5 ± 0.1*** |

(b) Task 3: comparison between the use of representations of different MLP layers to compute the Wasserstein.

| Labels | Accuracy ↑ | Fairness ↑ | DTO ↓ | Leakage ↓ |
|---|---|---|---|---|
| | | **Bios** | | |
| Representations | 82.4 ± 0.1 | **89.0 ± 0.3*** | **2.06*** | 96.5 ± 0.5 |
| Hard labels | **82.6 ± 0.2** | 87.5 ± 0.2 | 3.28 | **92.0 ± 0.2*** |
| | | **Moji** | | |
| Representations | **75.2 ± 0.1*** | **91.4 ± 0.3*** | **1.17*** | 86.9 ± 0.2 |
| Hard labels | 72.2 ± 0.1 | 65.0 ± 0.0 | 27.3 | **81.0 ± 0.8*** |

(c) Task 4: comparison between the use of representations $z_s$ and hard sensitive attributes to compute the Wasserstein distance.

## 6.4 Task 4: Independence with predicted hard sensitive attributes

We replace $z_s$ by the predicted $\hat{s}$ to compute the Wasserstein distance and report the results in Table 3c. We observe, on average, a slight improvement of the accuracy on Bios, and a slight decrease in accuracy on Moji. However, while the decrease in fairness is not significant for Bios, we observe a substantial drop for Moji. As a result, using $\hat{s}$ instead of $z_s$ seems to have a neutral impact at best, this may also result, in some cases, in a reduction of both accuracy and fairness.

## 7 Conclusion

We presented WFC a novel method that enforces fairness constraints using a pre-trained neural network on the sensitive attributes and Wasserstein regularization. Our model is theoretically well-motivated and has interesting properties over existing models. The most important one is the fact that it does not require annotation of the sensitive attribute at both training and inference time. We obtain competitive results compared to baselines on the Bios dataset and outperform them on the fairness score with comparable accuracy on Moji dataset. Furthermore, we present a solution for our algorithm to be trained when sensitive attributes are not available for a given dataset, paving the way for its use under realistic applications. In further studies, we will focus on applying this method using different text encoders or decoders, datasets, and downstream tasks, as this method can generalize to tasks out of the text classification scope, notably, regression and even unsupervised objectives.

## Limitations

The proposed approach is rather flexible as it can handle various types of sensitive attributes. However, due to the lack of available datasets, we were not able to assess our performance for continuous sensitive attributes, e.g. age. In addition, we are aware that gender may embrace a $n$-ary definition, in all our experiments, we were limited to consider-

ing only men vs women classification, due to data availability.

For Task 2 defined in the experiments section, we were able to show empirically that our method works well when the *demonic* model is pre-trained on a different dataset when no sensitive attributes or very few of them are available on the main training dataset. However, we do not provide sufficient generalization guarantees to consider an out-of-the-box large-scale deployment. The next step will be to derive theoretical guarantees inspired by the results of domain adaptation to assess how well this idea can be generalized to other data and under which conditions it might fail or succeed.

Finally, for Task 1 we did not perform a statistical test to assess the significance of the observed differences. Indeed, most of the results were reported from (Shen et al., 2022a) and we were unable to retrieve the scores for each run.

## Ethics Statement

We acknowledge the following concerns about our work. First, biases present in the data are US-centered, and concerning the Bias in Bios dataset genders are binary. Furthermore, to conduct our research we need to have access to the sensitive attributes contrary to what privacy measures recommend, and the latter are annotated with a risk of subjectivity.

## Acknowledgements

This work was funded by the french National Agency for Research (ANR) in the context of the Diké project (ANR-21-CE23-0026).
Our experiments utilize the previously mentioned Fairlib framework. We would like to express our gratitude to Xudong Han for his availability and assistance in using it.

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

# A Theoretical background and Proof

## A.1 Wasserstein Distance

Finding correspondences between two sets of points is a longstanding issue in machine learning. The optimal transport (OT) (Monge, 1781) problem offers an efficient solution to this issue by calculating an optimal one-to-one transport map between the two sets. This map is determined by considering the geometrical proximity of the

---

**Algorithm 1:** WFC Algorithm

---
**Data:** $D = \{(x_i, y_i, s_i)\}_{i=1}^n$ the training set, $n_e$ the number of epochs, $n_c$ and $n_d$ the number of training iterations per epoch for the critic and the classifier respectively, a batch size $n_b$, two neural networks $h_s(x)$ and $h_c(x; \theta)$, a Critic $C_\omega$, a weight on the regularization $\beta$

**for** *e = 1, ..., $n_e$* **do**

    **for** *t = 1, ..., $n_c$* **do**

        Sample $\{x_i, y_i, s_i\}_{i=1}^{n_b}$

        Encode : $z_s \leftarrow \{h_s(x_i)\}_{i=1}^{n_b}$, $z_y \leftarrow \{h_c(x_i)\}_{i=1}^{n_b}$

        Concatenate vectors to get $Z_{dep} \leftarrow [z_{s,i}, z_{y,i}]_{i=1}^{n_b}$

        Shuffle the $z_{s,i}$ vectors.

        Concatenate vectors to get $Z_{ind} \leftarrow [z_{s,i}, z_{y,i}]_{i=1}^{n_b}$

        $grad(w) \leftarrow \nabla_\omega \frac{1}{n_b}(\sum_{i=1}^{n_b} C_\omega(Z_{dep,i}) - \sum_{i=1}^{n_b} C_\omega(Z_{ind,i}))$

        $\omega \leftarrow Adam(\omega; grad(w))$

    **end**

    **for** *t = 1, ..., $n_d$* **do**

        Sample $\{x_i, y_i, s_i\}_{i=1}^{n_b}$

        Encode : $z_s \leftarrow \{h_s(x_i)\}_{i=1}^{n_b}$, $z_y \leftarrow \{h_c(x_i)\}_{i=1}^{n_b}$

        Concatenate vectors to get $Z_{dep} = [z_{s,i}, z_{y,i}]_{i=1}^{n_b}$

        Shuffle the $z_{s,i}$ vectors.

        Concatenate vectors to get $Z_{ind} = [z_{s,i}, z_{y,i}]_{i=1}^{n_b}$

        $\mathcal{L} \leftarrow \sum_{i=1}^{n_b} \mathcal{L}(y_i, h(z_{y,i}))$

        $\mathcal{L} \leftarrow \mathcal{L} + \beta(\sum_{i=1}^{n_b} C_\omega(Z_{dep,i}) - \sum_{i=1}^{n_b} C_\omega(Z_{ind,i}))$

        $\theta \leftarrow Adam(\theta; \nabla_\theta \frac{1}{n_b}\mathcal{L})$

    **end**

**end**

---

points in the sets. In practice, OT can be expressed as a problem of aligning two empirical distributions $\mathbb{P}_{X_1}$ and $\mathbb{P}_{X_2}$ supported on two point sets $X_1 = \{x_1^{(i)} \in \mathbb{R}^d\}_{i=1}^{N_1}$ and $X_2 = \{x_1^{(j)} \in \mathbb{R}^d\}_{i=1}^{N_2}$. We consider the Monge-Kantorovich formulation of this problem (Kantorovich, 1942) where the goal is to search for a coupling $\gamma$ defined as a joint probability distribution over $X_1 \times X_2$ with marginals $\mathbb{P}_{X_1}$ and $\mathbb{P}_{X_2}$. This amounts to minimizing the cost of transport w.r.t. some metric $l : X_1 \times X_2 \to \mathbb{R}^+$:

$$\min_{\gamma \in \Pi(\mathbb{P}_{X_1}, \mathbb{P}_{X_2})} \langle M, \gamma \rangle_F \qquad (11)$$

where $\langle \cdot, \cdot \rangle_F$ is the Frobenius dot product, $M$ is a dissimilarity matrix, i.e., $M_{ij} = l(x_1^{(i)}, x_2^{(j)})$, defining the cost of associating $x_1^{(i)}$ with $x_2^{(j)}$ and $\Pi(\mathbb{P}_{X_1}, \mathbb{P}_{X_2})$ is a set of doubly stochastic matrices. This problem admits a unique solution $\gamma^*$ and defines a metric on the space of probability measures called the Wasserstein distance (also known as the Earth-Mover Distance) as follows:

$$W_M(\mathbb{P}_{X_1}, \mathbb{P}_{X_2}) = \min_{\gamma \in \Pi(\mathbb{P}_{X_1}, \mathbb{P}_{X_2})} \langle M, \gamma \rangle_F.$$

### A.2 Proof that $\text{MI}(z_y, z_s) \geq \text{MI}(\hat{y}, \hat{s})$

We have :

$$p(\hat{y}, z_y, \hat{s}) = p(\hat{y}|z_y, \hat{s})p(z_y|\hat{s})p(\hat{s}) \qquad (12)$$

Recall that $\hat{s} = h_s(z_s)$ and $\hat{y} = h_c(z_y)$, therefore, $\hat{y}$ is fully determined by $z_y$, hence, $p(\hat{y}|z_y, \hat{s}) = p(\hat{y}|z_y)$. Then :

$$p(\hat{y}, z_y, \hat{s}) = p(\hat{y}|z_y)p(z_y|\hat{s})p(\hat{s}). \qquad (13)$$

Therefore, the three variables follow the markov property

$$\hat{s} \to z_y \to h_c(z_y). \qquad (14)$$

In such context, it was shown that (Yeung, 1991):

$$\text{MI}(z_y, \hat{s}) \geq \text{MI}(\hat{y}, \hat{s}). \qquad (15)$$

Following the same logic, we have

$$\text{MI}(z_y, z_s) \geq \text{MI}(z_y, \hat{s}) \qquad (16)$$

hence

$$\text{MI}(z_y, z_s) \geq \text{MI}(\hat{y}, \hat{s}). \qquad (17)$$

| Dataset | BiasInBios | Moji |
|---|---|---|
| input dimension | 768 | 2304 |
| hidden layers | 1 | 1 |
| hidden dimension | 300 | 300 |
| learning rate | 0.0001 | 0.00001 |
| batch size | 128 | 128 |
| epochs max | 10000 | 10000 |
| activation | TanH | TanH |
| $\beta$ | 1 | 1 |
| $n_c$ | 5 | 5 |
| $n_d$ | 20 | 5 |
| clamp value | 0.01 | 0.01 |
| layer used | last | last |

Table 4: Details on hyperparameters used for the classifying MLP.

| number hidden layer | 1 |
|---|---|
| hidden dimension | 512 |
| activation | ReLU |
| optimizer | Root Mean Square Propagation |
| learning rate | 5e-5 |

Table 5: Details on hyperparameters used for the Critic MLP.

## B    Experimental details

In this section, we provide additional experimental details, notably, we detail the MLP architectures, give the optimal hyperparameters, and describe the full algorithm of WFC.

### B.1    MLP architectures

In Table 4, we present the architectural details of the classifier MLP. We grid searched over the learning rate ($lr \in \{1e-5, 1e-4, 1e-3, 5e-5, 5e-4, 5e-3\}$, the number of training batches for classification per epoch $n_d \in \{5, 10, 20\}$, the value used to clamp the weights to enforce the Lipschitz constraint $clamp\ value \in \{0.001, 0.01, 0.1\}$, the parameter $\beta \in \{0.5, 1, 2\}$, the layer used between the *first hidden*, *last hidden*, or *last* layer.

### B.2    Critic architecture

In Table 5, we present the architectural details of the Critic, which is a simple multi-layer perceptron. We grid searched over the learning rate $lr \in \{5e-5, 5e-4, 5e-3\}$.

### B.3    Algorithm of WFC

In Algorithm 1, we provide the detailed algorithm for WFC used in our experiments.