# OpenReview forum: "Fair Text Classification with Wasserstein Independence"
_EMNLP/2023/Conference — EMNLP 2023 Main_

### Official Review · Reviewer_6C1r · 2023-08-01

**Soundness:** 3

**Excitement:**

3: Ambivalent: It has merits (e.g., it reports state-of-the-art results, the idea is nice), but there are key weaknesses (e.g., it describes incremental work), and it can significantly benefit from another round of revision. However, I won't object to accepting it if my co-reviewers champion it.

**Paper Topic And Main Contributions:**

The paper propose to improve fairness for text classification via minimizing the mutual information between the task representation and representation of the sensitive attribute. The mutual is estimated with the Wasserstein distance following previous works. Experiments show that the proposed objective is effective in improving fairness.

**Reasons To Accept:**

The paper is written clearly.

**Reasons To Reject:**

Since the objective is to minimize the mutual information, why adopting a low-bound for mutual information estimation? In [1], they also aim at improving fairness via minimizing mutual information, but adopt an upper-bound for estimation which is more plausible.

I think the novelty is limited, since [1] has proposed a similar idea and has a better methodology design (i.e., by using an upper-bound estimation). More importantly, [1] is neither compared or cited in the paper. The "Wasserstein Dependency Measure" in the paper follows previous works and is not appropriate to be used in this setting (as a lower-bound estimation).


[1] https://arxiv.org/pdf/2103.06413.pdf

**Reproducibility:**

3: Could reproduce the results with some difficulty. The settings of parameters are underspecified or subjectively determined; the training/evaluation data are not widely available.

**Reviewer Confidence:**

3: Pretty sure, but there's a chance I missed something. Although I have a good feel for this area in general, I did not carefully check the paper's details, e.g., the math, experimental design, or novelty.

---

> ### Author Rebuttal · Authors · 2023-08-28
>
> Authors : We thank you for your review and the interest you had in our paper.
>
> ***Reviewer : Since the objective is to minimize the mutual information, why adopting a low-bound for mutual information estimation? In [1], they also aim at improving fairness via minimizing mutual information, but adopt an upper-bound for estimation which is more plausible.***
>
> A : It seems that this remark may arise from a misunderstanding. As shown in equation (6), we do minimize an **upper bound** of the mutual information between the classification labels and the sensitive attribute, making the classification results independent to the sensitive attribute.
>
> ***R : I think the novelty is limited, since [1] has proposed a similar idea and has a better methodology design (i.e., by using an upper-bound estimation). More importantly, [1] is neither compared or cited in the paper. The "Wasserstein Dependency Measure" in the paper follows previous works and is not appropriate to be used in this setting (as a lower-bound estimation).***
>
> A : Regarding the reference you mentioned, thank you for pointing it; we will include it in the related work.
>
> According to our understanding, this paper aims to maximize the mutual information between pairs of representations of a sentence and its augmented version, varying according to the sensitive attribute *s*, and going through the same encoder, making the input *x* independent of *s*.
> However, this does not ensure independence between the prediction *ŷ* and *s* (cf. [1, 2]). In contrast, our theoretically grounded approach minimizes the mutual information between representations of the same sentence processed by two different encoders to make *ŷ* independent of *s*.
>
> Additionally, their approach relies on identifying biased attribute words, it is limited to the case where substitute words are available. This is a constraint we avoid.
>
> ---
>
> [1] Aili Shen, Xudong Han, Trevor Cohn, Timothy Baldwin, and Lea Frermann. 2022. Does Representational Fairness Imply Empirical Fairness?. In Findings of the Association for Computational Linguistics: AACL-IJCNLP 2022, pages 81–95, Online only. Association for Computational Linguistics.
> [2] Laura Cabello, Anna Katrine Jørgensen, and Anders Søgaard. 2023. On the Independence of Association Bias and Empirical Fairness in Language Models. In Proceedings of the 2023 ACM Conference on Fairness, Accountability, and Transparency (FAccT '23). Association for Computing Machinery, New York, NY, USA, 370–378.

---

### Official Review · Reviewer_nGpp · 2023-08-03

**Soundness:** 5

**Excitement:**

4: Strong: This paper deepens the understanding of some phenomenon or lowers the barriers to an existing research direction.

**Paper Topic And Main Contributions:**

This paper focuses on group fairness in text classification and proposes to improve fairness by introducing Wassertein independence. The proposed method is extensively examined over two standard fairness benchmark datasets, demonstrating compatible results compared to SOTA debiasing methods. More importantly,  the demonic model can be pre-trained and used for the debiasing purpose, enabling semi-supervised bias mitigation.

**Reasons To Accept:**

- The paper solves an important problem
- The paper is well-written and easy to follow
- The proposed approach is well-motivated and evaluated
- The proposed approach can be used without observing protected attributes during training
- The source code has been provided, which can be used in future research

**Reasons To Reject:**

- Many of the existing debiasing approaches do not need sensitive attributes at the inference time, including the baseline methods INLP, Adv, FairBatch, etc.

**Reproducibility:**

5: Could easily reproduce the results.

**Reviewer Confidence:**

4: Quite sure. I tried to check the important points carefully. It's unlikely, though conceivable, that I missed something that should affect my ratings.

---

> ### Author Rebuttal · Authors · 2023-08-28
>
> Authors : We genuinely value your thorough review and recognition of the problem's significance. Your acknowledgment of our efforts in evaluations and the advantage of a sensitive attribute-independent training method is appreciated. Below, we address your valid concern.
>
> ***Reviewer : Many of the existing debiasing approaches do not need sensitive attributes at the inference time, including the baseline methods INLP, Adv, FairBatch, etc.***
>
> Authors : The remark you made is common with other reviewer's, regrettably, this is a typographical error line 97 that has caused confusion for both you and R1. We sincerely apologize for any misunderstanding caused.
>
> Our approach, unlike current baselines, does not need sensitive attribute annotations in the training data, as detailed in line 420 (under "Training the demonic model") and empirically validated in Task 2 thanks to the code provided on GitHub.

---

### Official Review · Reviewer_Xr2p · 2023-08-05

**Soundness:** 4

**Excitement:**

3: Ambivalent: It has merits (e.g., it reports state-of-the-art results, the idea is nice), but there are key weaknesses (e.g., it describes incremental work), and it can significantly benefit from another round of revision. However, I won't object to accepting it if my co-reviewers champion it.

**Paper Topic And Main Contributions:**

This paper presents a training method for improving group fairness in text classification. Posing fairness as minimizing the mutual information between the label and the sensitive attribute represented in a piece of text (such as age, gender, or race), the authors propose to operationalize this optimization as minimizing the Wasserstein-1 distance between their representations' joint probabilities and product of marginals and implement it using Kantorovich-Rubinstein duality as in WGANs.

With experiments on two datasets (with gender and race as sensitive attributes), the paper shows experiments with two cases: one where the sensitive attribute is available at training time, and the other where it is not (in which case they train an attribute classifier on a separate dataset). They report fairness metrics such as GAP, DTO, and Leakage, and show that their proposed method overall matches performance of previous methods while being fairer in terms of GAP and DTO.

**Questions For The Authors:**

See weaknesses

**Reasons To Accept:**

1. Good motivation in using Wasserstein distance for improving fairness, as has been studied in image classification. It has shown to work better with GANs than adversarial learning as well so it makes sense to implement it for this setting.
2. The experiment setup is thorough with many different settings considered in terms of the availability of sensitive attributes, what "representations" to compute the loss on, and so on. Good fairness improvements on some of the datasets considered.

**Reasons To Reject:**

1. I struggled to draw clear conclusions from the experiments. Of the two datasets considered, while GAP and DTO are improved on one of them, the approach is not able to outperform many of the baselines (in accuracy or fairness). Further, leakage results are very very poor. The representations basically leak the sensitive attributes nearly perfectly in both sets of experiments. With the motivation of the method being minimizing dependence between the label and sensitive attribute, shouldn't leakage results be better? There is no discussion provided for this behavior either.

2. The paper makes a claim over and over that their method is applicable in cases where sensitive attributes might not be available at test time. To my knowledge, this is true for all the baselines they considered. Even if they were not available at train time, in most of the baselines considered, you can still train an attribute classification and apply their method the same as this paper does (as has been done in previous works like this: https://arxiv.org/abs/2307.13081). Using W-distance is orthogonal to this claim and the former seems to have mixed results anyway.

3. Many baselines related to INLP seem to be missing such as all the follow-up work from the same authors: https://proceedings.mlr.press/v162/ravfogel22a/ravfogel22a.pdf

**Reproducibility:**

4: Could mostly reproduce the results, but there may be some variation because of sample variance or minor variations in their interpretation of the protocol or method.

**Reviewer Confidence:**

4: Quite sure. I tried to check the important points carefully. It's unlikely, though conceivable, that I missed something that should affect my ratings.

---

> ### Author Rebuttal · Authors · 2023-08-28
>
> Authors : First of all, we thank you for the care and attention you have given to our document, and your appreciation of our efforts in testing various settings. Similarly, we thank you for pointing out the interest of using the Wasserstein distance to enhance model fairness.
>
> We also value your specific comments that will help in improving our work. Below, we provide clarifications to address some of your remarks and hope they answer your questions.
>
> ---
>
> ***Reviewer : I struggled to draw clear conclusions from the experiments. Of the two datasets considered, while GAP and DTO are improved on one of them, the approach is not able to outperform many of the baselines (in accuracy or fairness).***
>
> A : Indeed, our approach does not outperform all baselines on the Bias in Bios dataset (4th/9 in terms of accuracy fairness trade-off), but it does outperform baselines on the Moji dataset on both fairness and DTO (accuracy/fairness trade-off).
>
> Additionally, on top of being very competitive, our method has strong advantages over prior works:
> - it does not require annotation of the sensitive attribute at training time (matching GRPD regulations, for instance, that enforces more stringent requirements for the collection and utilization of protected attributes).
> - it can generalize to tasks out of the text classification scope.
> - it can be deployed in the case of continuous sensitive attributes.
>
> Those abilities afford a wide deployment of our approach in real-life scenarios and open the way to further work in this direction.
>
> ---
>
> ***R : Further, leakage results are very very poor. The representations basically leak the sensitive attributes nearly perfectly in both sets of experiments. With the motivation of the method being minimizing dependence between the label and sensitive attribute, shouldn't leakage results be better? There is no discussion provided for this behavior either.***
>
> A : While the method's motivation initially suggests better leakage, empirical fairness (Fairness) and representational fairness (Leakage) have been found to be uncorrelated and even opposing in various studies [1, 2]. We observe a similar phenomenon in INLP, which aims to remove sensitive information from the representations. Conversely, Condp significantly improves leakage but ranks poorly in terms of Fairness.
>
> Our primary goal is to improve empirical fairness while considering the trade-off with accuracy working on the representations. As you noted, we want to reduce the dependence between the label and sensitive attribute.
> However, optimizing for empirical fairness may not necessarily ameliorate representational fairness. Furthermore, reducing leakage makes it more challenging to retrieve sensitive attributes but could result in unintended information loss needed for the classification task without significantly enhancing empirical fairness, as previously mentioned.
>
> Still, we provide the leakage results for baseline comparison. Moreover, adjusting experiment parameters, such as the weight put on the Wasserstein regularization, can improve leakage. If we keep only the Wasserstein regularization in the loss, we can get a leakage of 53.9 for the Bias in Bios dataset and 77.1 for the Moji dataset.
>
> We conducted an extra experiment where we give more weight to the Wasserstein regularization in the loss. With a higher weight, the leakage decreases at the cost of accuracy. Considering $\alpha$ and $\beta$, respectively the weight on the classification loss and the Wasserstein regularization, we get the following results:
>
> | $\alpha$ | $\beta$ | DTO (Trade-off) | Accuracy | Fairness | Leakage |
> |---|---|---|---|---|---|
> | Bios |
> | 1 | 1 | 2.1 | 82.4 | 89.0 | 96.5 |
> | 1 | 5 | 2.3 | 81.8 | 89.3 | 87.4 |
> | 1 | 10 | 2.8 | 81.8 | 88.5 | 88.8 |
> | 1 | 20 | 3.6 | 81.3 | 88.0 | 81.8 |
> | Moji |
> | 1 | 1 | 1.2 | 75.2 | 91.4 | 86.9 |
> | 1 | 5 | 4.2 | 72.1 | 93.4 | 81.1 |
> | 1 | 10 | 5.8 | 70.5 | 92.1 | 81.6 |
> | 1 | 20 | 7.6 | 68.6 | 92.1 | 84.1 |
>
> On the Bias in Bios dataset, when leakage decreases, accuracy, that we want to preserve, immediately decreases and the trade-off worsen, even tho we note an improvement of empirical fairness. However, for a certain leakage both accuracy and fairness decrease, matching the observation of previously mentioned papers. On the Moji dataset, while we note an improvement of empirical fairness, the impact on accuracy is too strong to preserve a good trade-off.
>
> ---
>
> ***R : The paper makes a claim over and over that their method is applicable in cases where sensitive attributes might not be available at test time. To my knowledge, this is true for all the baselines they considered.***
>
> A : Thank you for pointing out this mistake, the claim in question appears in line 97. Regrettably, this is a typographical error that has caused confusion for both you and R2. We sincerely apologize for any misunderstanding caused.
>
> Our approach, unlike current baselines, does not need sensitive attribute annotations in the *training data*, as detailed in line 420 (under "Training the demonic model") and empirically validated in Task 2 using the code provided on GitHub.
>
> ---
>
> ***R : Even if they were not available at train time, in most of the baselines considered, you can still train an attribute classification and apply their method the same as this paper does (as has been done in previous works like this: https://arxiv.org/abs/2307.13081). Using W-distance is orthogonal to this claim and the former seems to have mixed results anyway.***
>
> Thank you for this additional reference that we can add to our related work. We were not able to acknowledge it since it was published after the submission deadline (24th of July only available on Arxiv).
>
> However, as the authors emphasize, employing proxy-sensitive attributes often worsens the fairness-accuracy trade-off, a key problem we address.
>
> Furthermore, this extension, although unrelated to the use of the Wasserstein distance as you mentioned, yields similar results to our original approach (Table 2.a) with additional benefits mentioned in the answer of the first remark. Thanks to this addition and unlike other baselines, we propose a version of our method compatible with recent regulations on the use of sensitive data.
>
> ---
>
> ***R : Many baselines related to INLP seem to be missing such as all the follow-up work from the same authors***
>
> A : We appreciate the recommendation of additional baselines and acknowledge that there may be more options due to the extensive work in the field. In our experiments, we made efforts to compare our work with eight baselines, including recent ones from 2022, following established protocols used in ACL conference papers (e.g., [1]).
>
> ---
>
> [1] Aili Shen, Xudong Han, Trevor Cohn, Timothy Baldwin, and Lea Frermann. 2022. Does Representational Fairness Imply Empirical Fairness?. In Findings of the Association for Computational Linguistics: AACL-IJCNLP 2022, pages 81–95, Online only. Association for Computational Linguistics.
> [2] Laura Cabello, Anna Katrine Jørgensen, and Anders Søgaard. 2023. On the Independence of Association Bias and Empirical Fairness in Language Models. In Proceedings of the 2023 ACM Conference on Fairness, Accountability, and Transparency (FAccT '23). Association for Computing Machinery, New York, NY, USA, 370–378.

---

### Meta-Review · Area_Chair_Vts5 · 2023-09-20

**Recommendation:** 5

**Metareview:**

This paper presents an approach to mitigating bias in text classifiers by inducing Wasserstein independence between representations learned to predict target labels and sensitive attributes. The paper is well-motivated and the experiment setup is thorough. The rebuttal addresses most of concerns.

---

### Decision · Program_Chairs · 2023-10-07

**Decision:**

Accept-Main

**Comment:**

This paper presents an approach to mitigating bias in text classifiers by inducing Wasserstein independence between representations learned to predict target labels and sensitive attributes. The paper is well-motivated and the experiment setup is thorough. The rebuttal addresses most of concerns.